# RevisedMedYOLO: Unlocking Model Performance by Careful Training Code Inspection

**Kai Geissler**[1]                                KAI.GEISSLER@MEVIS.FRAUNHOFER.DE
**Jan Hendrik Moltz**[1]
**Hans Meine**[1]
**Markus Wenzel**[1,2]

[1] *Fraunhofer Institute for Digital Medicine MEVIS, Bremen, Germany*

[2] *Constructor University, Bremen, Germany*

**Editors:** Accepted for publication at MIDL 2025

## Abstract

The MedYOLO architecture, adapting YOLOv5 for 3D medical object detection, was reported by its original authors to have failed to train effectively in LIDC for the detection of lung lesions and to fail in BraTS for the detection of brain tumors when using a large model configuration. This work introduces RevisedMedYOLO[1], achieved by carefully reviewing and correcting the original training implementation. We fixed critical bugs related to dataset shuffling, initialization of bias parameters, and bounding box clamping during zoom augmentation. Consequently, RevisedMedYOLO demonstrates successful learning on these datasets (AP@0.5 > 0), unlike the original implementation. This study underscores the crucial role of careful code implementation and debugging in enabling deep learning model performance for challenging tasks in medical image analysis.

**Keywords:** MedYOLO, Medical Object Detection, Lesion Detection, 3D Object Detection

## 1. Introduction

The YOLO family of deep learning models (Redmon et al., 2016) is a well-known family of models for object detection in 2D images. Two-dimensional YOLO-based models have been successfully applied in a wide range of applications in the medical domain (Ragab et al., 2024). In a recent study Sobek et al. (2024) extended YOLOV5 (Jocher, 2020) to object detection in 3D medical images, calling it MedYOLO. The benefit of MedYOLO compared to other medical object detection models such as nnDetection (Baumgartner et al., 2021) is that its light weight model design and one-shot architecture make it very computationally efficient. MedYOLO has been reported to work well for the detection of abdominal organs and the detection of heart and thoracic aorta in CT scans. However, the MedYOLO authors reported that the model did not work at all for the detection of lung lesions in the LIDC dataset (Armato III et al., 2015) and worked only for one of the two model configurations tested for the detection of brain tumors in the BraTS 2021 dataset (Baid et al., 2023). In this study, we present a revised version of MedYOLO. For this, we performed a thorough code review and refactored the code where needed, which enabled the model to work properly in experiments on which it has been reported to fail before.

---

1. Code available at: github.com/FraunhoferMEVIS/RevisedMedYOLO

## 2. Methods

Our changes to the MedYOLO training setup were as follows. First, we enabled dataset shuffling for the training data, as this is important to ensure unbiased gradient estimates for each iteration of stochastic gradient descent. Especially if the dataset is ordered with respect to some data characteristic, training with unshuffled data is highly detrimental to the training process. Second, we fixed an index error in the weight initialization of the objectness neuron bias. In YOLOV5 this is initialized to a small value that depends on the image size, so that the model defaults to predicting most of the image as background. When moving from 2D to 3D the indexing of the bias initialization was not properly adapted, and therefore the model initialized the bias weight of the bounding box widths to extremely small values, while the objectness bias was not initialized to small values accordingly. This had the consequence that the initial model predicts extremely narrow bounding boxes in one dimension and predicts too much foreground, so the model training always first has to recover from this improper initialization. Third, in the random zoom augmentation, we fixed an off-by-one error in the clamping of bounding box coordinates for zoom factors greater than one. As MedYOLO uses only cubic images, this only caused the height of the boxes not to be clamped correctly, potentially leading to box heights that reached out of the visible image area.

After applying these fixes, we retrained the model on the two public datasets used by Sobek et al. (2024), which were LIDC and BraTS. We split the data into training, validation and test data using a 70/10/20 split, resulting in 704/101/201 training/validation/test cases for LIDC and 876/125/250 for BraTS. We used the validation data to select the best model during training, which was later evaluated on the test split. The models for both datasets were trained using an image size of $350 \times 350 \times 350$. In the LIDC data, only the small MedYOLO configuration was trained, while on BraTS both a small and a large variant were trained. The large variant uses 64 filters in the first convolutional layer with width and depth multipliers of 1, while the small variant uses a width multiplier of 0.5 and a depth multiplier of 0.33, significantly reducing the model capacity. To stay close to the original MedYOLO study, we used the same hyperparameters and also report the average precision metrics at a bounding box overlap of 0.5 (AP@0.5) and additionally at an overlap of 0.1 (AP@0.1).

## 3. Results

Our results are shown in Table 1. Although the original MedYOLO did not learn properly in LIDC, our adapted model achieves an AP@0.5 of 0.102. This result is not very high, but suffices to show that the model can in principle be trained successfully on the dataset. Also note that LIDC is a challenging dataset because the lung lesions are mostly very small and therefore difficult to detect and localize. We assume that by proper hyperparameter tuning these results can be significantly improved. In BraTS Sobek et al. (2024) reported an AP@0.5 of 0.861 for the small model and 0.000 for the large model. With our adaptions, the model achieves an AP@0.5 of 0.747 for the small model and 0.008 for the large model. It is important to point out that we used different data splits than the original MedYOLO study because we do not know which ones they used. Therefore, the metric for the small model in BraTS is only meant to show that our model performance stays in the same range and

Table 1: Average precision at a bounding box overlap threshold of 0.5 (@0.5) and 0.1(@0.1) for selected datasets. MedYOLO results shown as reported in Sobek et al. (2024); RevisedMedYOLO shows our results.

| Dataset (Model) | MedYOLO@0.5 | RevisedMedYOLO@0.5 | RevisedMedYOLO@0.1 |
|---|---|---|---|
| LIDC (Small) | 0.000 | 0.102 | 0.361 |
| BraTS (Small) | 0.861 | 0.747 | 0.813 |
| BraTS (Large) | 0.000 | 0.008 | 0.411 |

should not be used to judge which model is better. Noting the large discrepancy between the small and large model versions, we inspected the training and validation loss curves and observed that the training loss decreased steadily over time while the validation loss plateaued. We hypothesize that the large MedYOLO model overfits heavily on the BraTS data, causing the reduced model performance compared to the small model.

In addition, we note that average precision at a bounding box overlap of 0.5 is not well suited for 3D object detection. For 3D boxes, this is already a quite large overlap because the additional third dimension adds into the overlap error. Therefore, many model predictions which capture the reference object quite well are considered false positives in the metric computation. When we instead compute the average precision at a bounding box overlap of 0.1, the result on LIDC increases from 0.102 to 0.361, and the BraTS results increase from 0.747 to 0.813 and from 0.008 to 0.411, indicating that the models detect many more objects correctly than indicated by the small AP@0.5 values.

To summarize our observations, we find that models such as RevisedMedYOLO are much more broadly applicable to 3D medical object detection than it appeared in the original MedYOLO study (Sobek et al., 2024). It will be the subject of future work to thoroughly evaluate the best achievable performance of RevisedMedYOLO and compare it with other established 3D medical object detection methods, such as nnDetection.

## 4. Conclusion

We revised MedYOLO, an object detection model for 3D medical images, by carefully inspecting its training code and fixing crucial issues. After the original model failed on two out of three experiments on public data, our RevisedMedYOLO works in all of these cases. We expect the training results to improve significantly from the current baseline by hyperparameter tuning and the addition of more data augmentation techniques, such as random image rotations and intensity augmentations.

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
