# OpenReview forum: "RevisedMedYOLO: Unlocking Model Performance by Careful Training Code Inspection"
_MIDL.io/2025/Short_Papers — MIDL 2025 - Short Papers_

### Official Review · Reviewer_691Y · 2025-04-23

**Rating:** 4
**Confidence:** 4

**Summary:**

The authors revise the training implementation of the MedYOLO architecture and correct three serious issues with the code, reporting an increased performance for datasets where the original implementation failed.

**Strengths:**

There is great importance of the work in assessing and improving already existing model architectures. The authors also propose more suitable evaluation metrics.

**Weaknesses:**

The decreased performance for their small model trained on BRaTS is attributed to a different data split, however wouldn't it be possible that the increased performance is also attributed to such changes? The difference between 0.861 and 0.747 is more significant than the difference between 0.000 and 0.008. It is also unclear how well the original model performs with respect to the new, more suitable evaluation metric. Perhaps it achieves better results, despite achieving an overlap threshold of 0.000.

---

### Decision · Program_Chairs · 2025-05-01

Accept